# Monitoring of Ship Operations in Seaport Areas in the Sustainable Development of Ocean–Land Connections

Teresa Abramowicz-Gerigk [1,*], Zbigniew Burciu [1], Miroslaw K. Gerigk [2] and Jacek Jachowski [1]

1    Faculty of Navigation, Gdynia Maritime University, 81-225 Gdynia, Poland; z.burciu@wn.umg.edu.pl (Z.B.); j.jachowski@wn.umg.edu.pl (J.J.)
2    Faculty of Mechanical Engineering and Ship Technology, Gdansk University of Technology, 80-233 Gdansk, Poland; mger@pg.edu.pl
*    Correspondence: t.gerigk@wn.umg.edu.pl; Tel.: +48-585-586-158

**Abstract:** The paper is devoted to underlining the important role of monitoring systems in the sustainable development of seaport areas—sensitive ocean–land connections exposed to the harmful effects of multimodal transport. The study concerns the existing monitoring possibilities of the environmental factors and ship traffic near port infrastructure. The main aim of the study is presenting the example of solutions, supporting the sustainable development of port areas, related to the most dangerous ship maneuvering operations carried out near the berths. An indirect method for measuring loads on the seabed from the propeller and thruster jets during ship berthing and an experimental method for predicting the hydrodynamic forces generated on a moored Panamax-size bulk carrier by a similar vessel passing along in shallow water conditions are described in the context of their implementation in monitoring systems. The cloud-based system—installed in the ferry terminal in the Port of Gdynia and developed for monitoring the flow generated by the ship propellers during maneuvers near the berth and warning about the exceedance of allowable pressure on the quay wall—allows, after a two-year operation, to draw the conclusions related to maintenance planning and has an impact on port sustainability. The discussion presented in the paper underlines the influence of monitoring both the environmental elements and hazardous ship operations on the sustainable development of seaport areas.

**Keywords:** seaport; cloud-based monitoring system; ship berthing; moored ship; passing ship

## 1. Introduction

The sustainable development of seaport areas—sensitive connections between ocean and land, largely exposed to the negative effects of multimodal transport—focuses on limiting the impact of ship activities on port infrastructure and environment, minimizing economic and social costs and decreasing health risks to residents of coastal areas.

The importance of monitoring environmental elements and the influence of ship operations on port infrastructure has increased with the growing number of large seagoing vessels. The economy of scale and energy efficiency are the fundamental principles of the economics of maritime transport. The larger the ship, the lower the cost per unit transported. However, the number of ports capable of handling large ships is limited, and, consequently, many ports operate close to their operational limits. The advantages of size, most noticeable for liner shipping of ultra-large container ships (ULCS) with capacities of 18,000–25,000 TEU (Twenty-Foot Units), yield negative returns during port operations, involving much higher costs to maintain an acceptable level of service [1].

A large number of containers arriving at the same time is challenging for multimodal transport, i.e., maritime, road and rail transport in the coastal area. Trucks with containers cause more air pollution than a large ro-pax, ro-ro ferry or container ship.

Bernacki [2] discussed the economy of scale and decrease in transport costs versus adapting a port to handle large ships by dredging a fairway and port docks. Dredging,

which largely determines the accessibility of ports, is costly and harmful to the environment and should be limited to a minimum. This means that when adapting the port to operate larger ships, we should leave a minimum safety margin.

Lee et al. [3] presented a literature study on economic, environmental and social dimensions of sustainable development in maritime transport, concluding that the introduction of monitoring systems in ports contributes to better planning of port operations and new investments, taking into account the port sustainability and reduction of renovation and maintenance work.

Air and water pollution affect the quality of life of people living in coastal areas. Most European ports, which experience operational challenges related to climate change, consider the impact of these changes in their development projects [4]. The emission reduction policies are based on the factors influencing ship emissions, including monitoring berth availability and accessibility, which affects the level of ship emissions [5] performed by maritime authorities. Lim et al. [6], based on the systematic literature review of port sustainability, proposed the analytical port sustainability indicators and concluded that the future research should encourage systematic improvements in the port operational practice.

The monitoring of environmental elements varies widely as weather conditions determine port operational windows and should be considered in planning ship maneuvering and loading operations. The monitoring of the ship operations' impact on marine structures is less frequently used. Usually, it is related to berthing energy control, mostly in gas and oil terminals servicing large tankers, where docking systems significantly decrease the risk of pollution as a result of an accident during berthing.

Various systems are being installed in seaports to monitor the strength parameters of quays and determine their durability, since the condition of port infrastructure has a direct impact on ship operations and environment. The importance of remote strength monitoring increases if a quay needs to be taken out of service for the duration of an inspection. The remote measurements are a valuable source of online information about the condition of the structure between inspections [7,8].

A much less recognized problem is the monitoring of seabed scouring near the berth by ship propeller jets and its effect on the berth stability, because the measuring elements cannot be installed on the seabed. The paper presents an approach used in the monitoring of jet velocities over the seabed based on the indirect method of pressure measurements on the quay wall using CFD simulation [8,9].

The determination of the hydrodynamic forces generated by a passing ship on a moored ship—which cause the moored ship to come into contact with the quay wall and the fendering and mooring systems—based on scale model tests in deep water conditions, presented in [10], showed a significant influence of the passing ship velocity and distance on the moored vessel. The new results of tests in shallow water presented in the article indicate that the forces in shallow water are twice as high as in deep water. The new results are compared to the existing empirical methods [11,12].

The approach that is developing the fastest in port operations is the application of smart port solutions composed of intelligent technologies and advanced digitization and communication technologies. The most advanced application is a digital twin of the whole port based on sensor networks, UAVs (unmanned aerial vehicles) and smart cameras, which give real-time information about activities inside the port and the current state of port infrastructure. This application was introduced in the Port of Antwerp [13].

The cloud-based monitoring, allowing for compilation of several monitoring systems and the introduction of digital twins of marine constructions in the port, is discussed in the paper in relation to the main factors of port sustainability.

The structure of the presented paper is as follows: the background of the study presented in the Introduction is followed, in the second paragraph, by examples of numerical and experimental investigations carried out using large-scale manned models, allowing for a prediction of ship impact on port infrastructure and the development of monitoring systems. In the third paragraph, the experimental data of passing ship effect are verified

with the data available in the literature, showing the strong influence of water depth to ship draft ratio on the coefficients of hydrodynamic forces generated on the moored ship. The fourth paragraph—Discussion—shows the advantages of the presented investigations in the case of using them in monitoring systems. Based on the authors' experience, examples of monitoring systems used in the biggest European ports and the literature study the influence of the monitoring of ship operations on the sustainable development of seaports is summarized. In the fifth paragraph—Conclusions—the prognosis of future smart port development is discussed.

## 2. Prediction and Monitoring of Loads Generated by Ships on the Berth and Seabed near the Quay Wall

### 2.1. Prediction and Monitoring of Thruster and Propeller Jets' Scouring Effects

The prediction of thruster and propeller jets' scouring effects can be based on empirical methods or the CFD simulation. An example of the prediction of the influence of bow thruster jets on the seabed protection in the ro-pax ferry terminal in the Port of Gdynia, Poland, is presented in Figure 1. The hull form model developed for the 160 m long ro-pax ferry and the grid of pressure sensors arranged on the model of the quay wall are presented in Figure 1a. The computing domain with overlapping meshes and an image of the velocity field on the quay wall and on the seabed for bow thrusters operating for 10 s with thrusts equal to 130 kN are presented in Figure 1b,c, respectively [9].

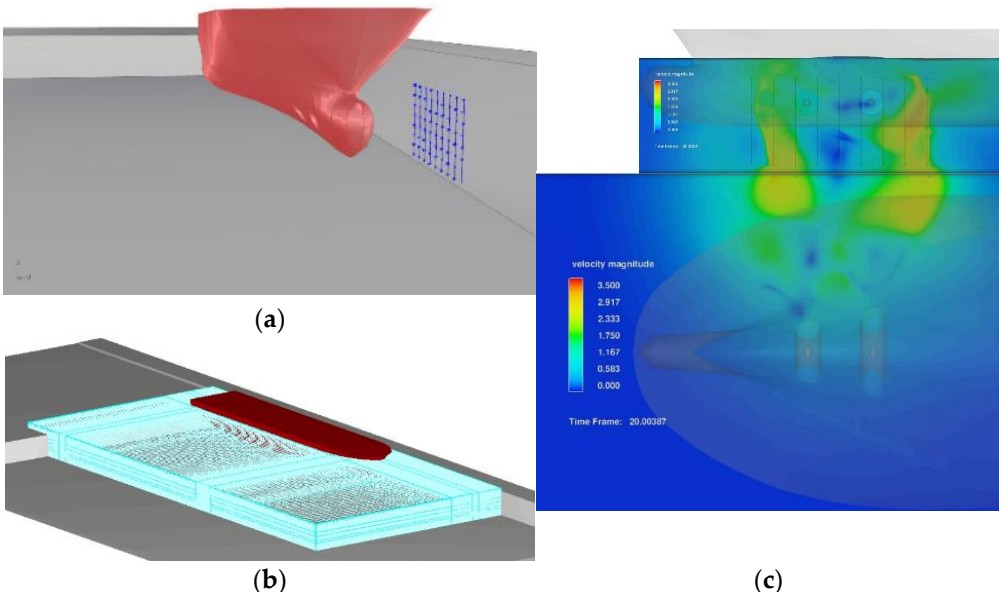

(a)

(b)

(c)

**Figure 1.** Indirect method for prediction of bow thruster jet velocities over the seabed: (**a**) hull form model of the ferry and grid of pressure sensor positions on the quay wall; (**b**) computing domain with overlapping meshes; (**c**) CFD simulation results: image of the velocity field on the quay wall and on the seabed for both bow thrusters operating for 10 s with thrusts equal to 130 kN. (Developed on the basis of [9]).

The measuring system installed on the quay wall in front of the bow thruster position of the moored ferry is presented in Figures 2 and 3. The supporting structure for installation on the quay wall with pressure meters is presented in Figure 2. The drawing of the quay wall with the measuring system and layout of the seabed protection mattress elements along the quay wall are presented in Figure 3a [14]. The 3D distribution of the bow thruster-induced pressure field measured on the quay wall during the unberthing maneuver in the 7th second of bow thruster operation is presented in Figure 3b [14].

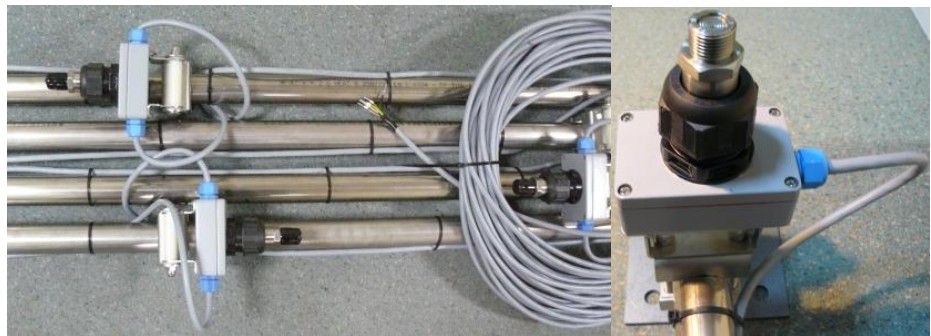

**Figure 2.** Measuring system of bow thruster jet-induced pressure on the quay wall: elements of the measuring system with pressure meters.

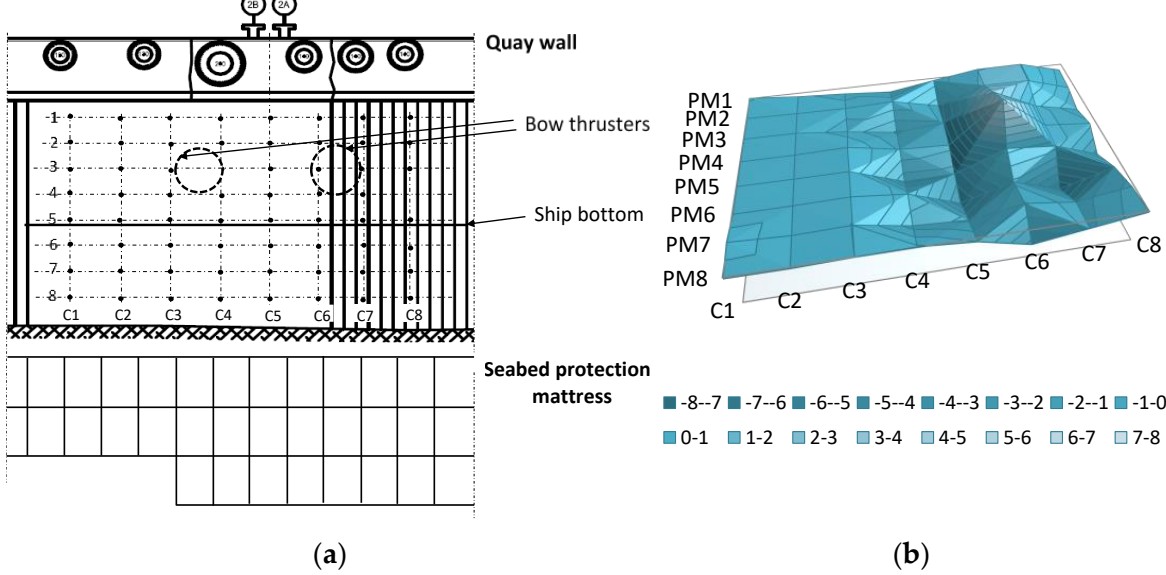

(**a**) (**b**)

**Figure 3.** Measuring system of bow thruster jet-induced pressure on the quay wall: (**a**) view on the quay wall with the measuring system and layout of seabed protection mattress elements; (**b**) 3D distribution of the bow thruster-induced pressure field [kPa] measured on the quay wall during the unberthing maneuver in t = 7 s of the bow thruster operation. (Developed on the basis of [14]).

The similar method based on the measurements of pressure generated by propellers on the quay wall was applied in the cloud-based monitoring system of loads generated by propeller jets near the ramp. The E-QuayTracker system has operated in the ferry terminal in the Port of Gdynia for two years, allowing for the online observation of pressure generated on the quay wall near the ramp by berthing ro-pax ferries. The functional modules, hardware and software characteristics of the system are described in [8].

The scheme of the system architecture is presented in Figure 4.

The open architecture of the system allows for the simultaneous monitoring of many berths and other marine structures and integration of this system with other port monitoring systems. Two-year operation of the system prototype allowed the authors to draw conclusions related to its implementation in the port with respect to the technical and human factor aspects.

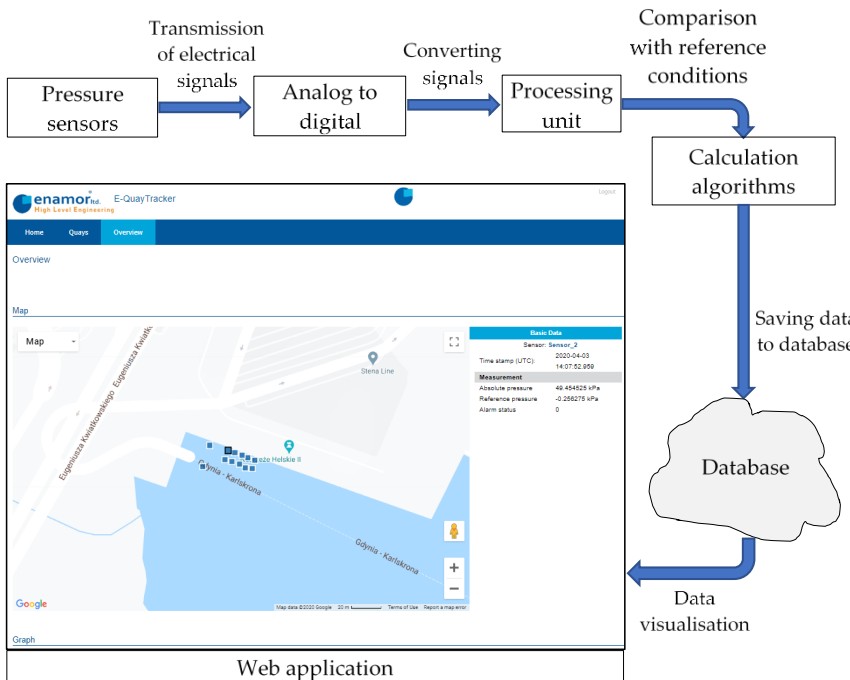

**Figure 4.** Architecture of the cloud-based monitoring system E-QuayTracker for prediction of propeller loads on the seabed near the ramp.

### 2.2. Prediction of Passing Ship Generated Loads on the Moored Ship

The passing ship moving alongside the moored ship generates the flow field due to the hydrodynamic interactions. The interaction forces depend on the moored and passing ships characteristics, berth structure, fendering and mooring systems and water depth to ship draft ratio. The main operational parameters that can be controlled during the maneuvering operations are the separation distance and the velocity of the passing ship. The influence of weather conditions is most important for ships with large windage areas, but the passing ship's effect should be considered in all cases.

Predicting the interaction forces is necessary for berthing facility design and the development of traffic safety measures. The passing ship may interrupt or disrupt loading operations or even cause the moored ship to break from its moorings [15,16]. The empirical methods are based on the CFD simulation and model tests.

Most of the published results are based on small models' captive tests carried out in towing tanks in 1:135 [11], 1:100 [12,17], 1:68 [18], 1:60 [19] and 1:32 [20] model scales.

The tests carried out using small captive models do not allow for the examination of all the phenomena related to fluid flow in real scale. The flow blockage is the effect of small hydraulic clearance. This, in turn, causes water cushion formations. The areas of flow separation are also not exactly the same because of the Reynolds number—they are much greater in real scale. The proposed prediction method, using large, manned models of Panamax-size bulk carriers, reproduces the real maneuvering procedure with the free-floating, self-propelled passing ship. The results of the experiments carried out in deep water conditions have already been presented in [10].

The results in deep water conditions showed hydrodynamic force values more than twice as low as those in shallow water conditions.

The new investigations of passing ship effect were conducted at the Ship Handling Research and Training Centre (SHRT Centre) located on Lake Silm in Kamionka, near Ilawa, Poland. The experimental test setup was constructed in a shallow water area with a depth-to-ship draft ratio of 1.3. The experimental facilities with the quay wall mock-ups in deep and shallow water conditions are presented in Figure 5.

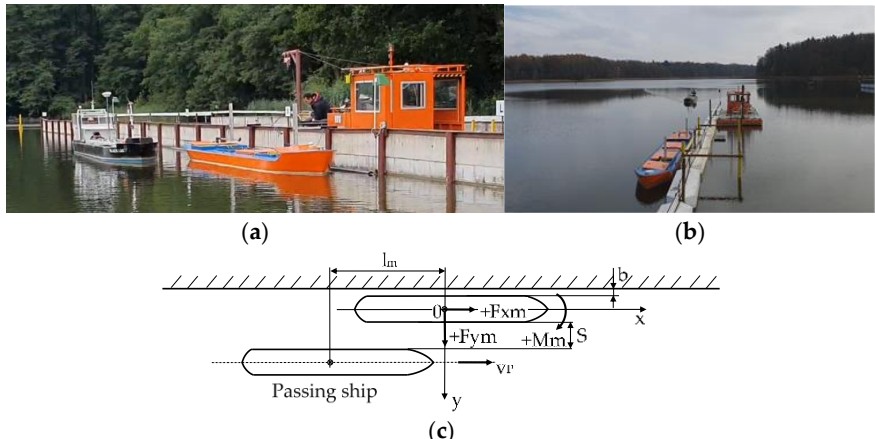

**Figure 5.** The experimental setups at the SHRT Centre on Lake Silm: (**a**) tests in deep water conditions [10]; (**b**) tests in shallow water conditions; (**c**) coordinate system adopted during the research [10].

The main ship models' parameters are presented in Table 1.

**Table 1.** Main parameters of moored and passing ship models.

| Dimension [m] | | Moored Ship Model | Passing Ship Model |
|---|---|---|---|
| L | Length overall | 9.49 | 9.17 |
| $L_{BP}$ | Length between perpendiculars | 9.00 | 9.30 |
| B | Breadth | 1.26 | 1.34 |
| T | Draft | 0.51 | 0.52 |

The geometrical model scale was 1:24. The berth mock-up structure is shown in Figure 6.

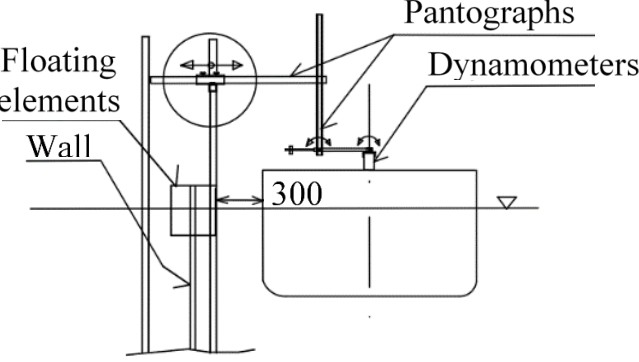

**Figure 6.** The berth mock-up structure.

The measuring method and instrumentation were described in [10].

## 3. Results of the Experimental Study on Passing Ship-Generated Forces on the Moored Ship in Shallow Water Conditions

### 3.1. Results of the Model Tests

The program of trials included five passing velocities at each of the three assumed separation distances, "s", equal to 1, 2 and 2.5 moored ship model breadths.

The tested passing ship velocities converted to the real scale was in the range of 3 knots to 12 knots (1.7 m/s to 6 m/s). An example of measured transverse force (sway force) and longitudinal force (surge force) converted to the real scale Fy and Fx, respectively,

dependent on the non-dimensional distance "l" between the ships, defined in Figure 5c and related to the length of the moored ship, for the separation distances s = 1 and s = 2, is presented in Figures 7 and 8.

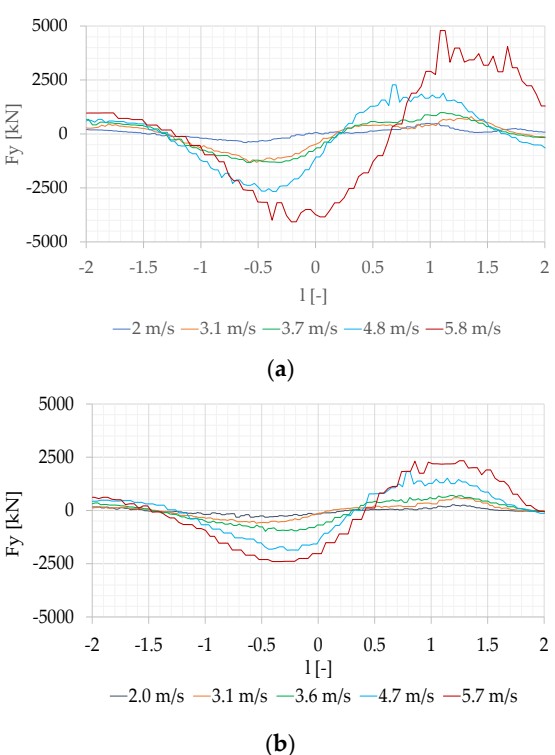

**Figure 7.** Sway force Fy: (**a**) separation distance s = 1; (**b**) separation distance s = 2.

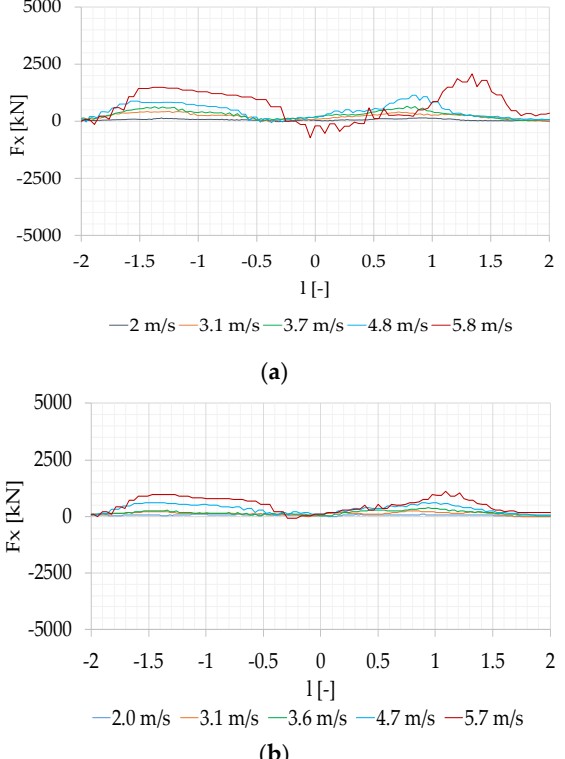

**Figure 8.** Surge force Fx: (**a**) separation distance s = 1; (**b**) separation distance s = 2.

The real scale values were calculated using the Froude law of similarity according to Equations (1)–(3).

$$Fy = Fym \cdot \lambda^3, \tag{1}$$

$$Fx = Fxm \cdot \lambda^3, \tag{2}$$

where Fy and Fx are the transverse and longitudinal forces generated on the moored ship in real scale, Fym and Fxm are the forces on the model, respectively, and λ is the geometrical scale factor.

$$v = v_P \sqrt{\lambda} \tag{3}$$

where v is the passing ship velocity, and $v_P$ is the passing model velocity.

### 3.2. Verification of the Results of the Model Tests

In the considered case of similar moored and passing ships' sizes, the hydrodynamic forces are strongly dependent on the combination of passing distance and passing ship velocity. The speed of the passing ship cannot be too low because the ship may have poor control over its movement. From the point of view of navigation safety and efficient traffic management, maximum speed reduction is not always a desirable option [16].

To compare the obtained results with the results presented in the literature [11,12], the non-dimensional coefficients of surge and sway forces and non-dimensional passing ship velocity Fn (Froude number) were used in the form of Equations (4)–(6), respectively [10]:

$$Cx = Fx/0.5 \cdot \rho v^2 LT, \tag{4}$$

$$Cy = Fy/0.5 \cdot \rho v^2 LT, \tag{5}$$

$$Fn = v/\sqrt{g \cdot L} \tag{6}$$

where Cx is the longitudinal force coefficient, Cy is the transverse force coefficient, $\rho$ [kg/m$^3$] is the water density and T [m] and L [m] are the moored ship dimensions.

The obtained maximum negative and positive Cx and Cy values, e.g., Cxmin, Cxmax, Cymin and Cymax, at v = 3 m/s to 6 m/s compared to the results published in [11] for the ship moored at the short tight wall, averaged velocities of v = 3 m/s to 6 m/s at h/T = 1.49, 1.32, 1.22 and 1.11, and the results published in [12] for the ship moored at the long tight wall and passing ship velocity v= 4 m/s, h/T = 1.2 are presented in Figure 9.

The presented model test results are generally close to the results of the 1:135 [11] and 1:100 [12] model scales. However, the coefficients Cxmin and Cxmax are generally smaller, Cymin is the same or bigger and Cymax is generally smaller than that presented in [11,12]. The strong dependence of shallow water is for h = 1.11, where the Cymax coefficient is several times bigger than at h = 1.22, 1.3, 1.32 and 1.49. It is related to the strong shallow water effect with the different phenomena.

The differences between the results of the presented field experiments and captive tests are also related to the natural conditions of the experimental environment, with the water depth to draft ratio of h/T = 1.3 at the wall and possible small depressions in the exercise area. The bigger Cxmax and Cymax values presented in [11] are mainly related to the short wharf, with the length not much bigger than the ship length.

The presented results can be used to determine the limiting operational values of the speed and velocity of the passing ship. The maximum measured sway force was generated by the passing vessel at 12 knots at the smallest separation distance equal to the ship breadth. This force was equal to 5000 kN.

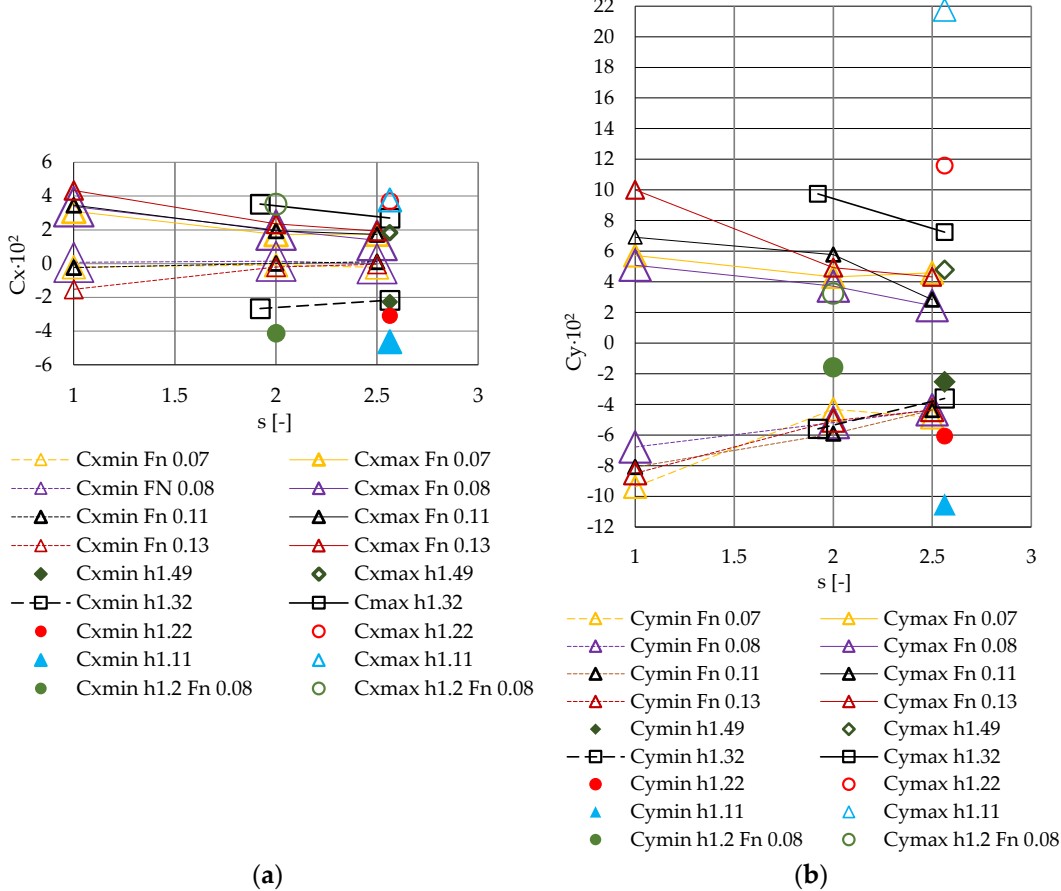

**Figure 9.** Coefficients of hydrodynamic forces: (**a**) Cxmin and Cxmax, (**b**) Cymin and Cymax; measured at v = 3 m/s to 6 m/s, marked as Fn 0.07, Fn 0.08, Fn 0.11, Fn 0.13; the results published in [11] for ship moored at short wharf averaged, for velocities, v = 3 m/s to 6 m/s (Fn = 0.06 to Fn = 0.16) at h/T = 1.49, 1.32, 1.22 and 1.11, marked as h1.49, h1.32, h1.22, h1.11; the results published in [12] for ship moored at the long tight wall v= 4 m/s, h/T = 1.2, (Fn = 0.08) marked as h1.2 Fn 0.08; s—non-dimensional distance between the vessels related to the moored ship breadth.

The maximum allowable loads on ship bollards are equal to 500 kN; therefore, the configuration of the three headlines, bow spring, stern spring and three stern lines usually used for mooring a Panamx-size bulk carrier allow for safe loading operations. However, in strong weather conditions, this number of mooring lines is not sufficient.

From an operational point of view, the maximum passing velocity in most cases should be no greater than 5 knots, and the maximum allowable ship motion amplitudes for bulk carriers should be about 1 m. It was confirmed in the presented study that even at small separation distances, the mooring forces at a passing velocity of about 5 knots would not be greater than 1000 kN. However, under strong winds, higher velocity values could be used by captains to keep a straight course; therefore, the monitoring of passing ships would be important for port traffic services, especially in bad weather conditions.

## 4. Discussion

Two years of monitoring berthing operations at the ferry terminal in the Port of Gdynia showed the possibility of online detection of the operational limit exceedance. The presence of the monitoring system helped highlight the influence of human factors on berthing operations and increase captains' concentration during maneuvers. The captains were more attentive to the port's recommended limits on propulsion power used at the quay [1].

The fuzzy model for the determination of the maximum hydrodynamic forces generated by the passing ship on the moored ship, presented in [21], is an example of an

application of the presented experiments in the creation of the control system for autonomous navigation. These results can be also used in traffic monitoring systems, advising the safe passing distance and speed.

The limitations of the presented study are related to the scope of the presented research, including only the selected cases of the wide range of port operations. Two examples of prediction and monitoring of the ship operations, which have an impact on the port infrastructure, preventing disruptions in port operation and reducing the $CO_2$ footprint, are between different solutions used in sustainable seaports.

The monitoring systems protecting against the structural instability and failure of marine structures decrease the carbon footprint of maneuvering operations. The frequent operation of high-powered, self-berthing ro-ro and ro-pax vessels has an impact on the environment due to the necessity of the installation of scour protection structures, their surveys and their maintenance work, which are dependent on the performance of ship operational procedures [22,23].

In the case of advanced smart port solutions, the processed real-time information of activities inside the port and the current state of port infrastructure are combined and can be used in a digital twin of the entire port, as for the development of the digital twin in the Port of Antwerp, for instance [13]. The monitoring systems used in the Port of Antwerp are a traffic monitoring and management system based on digital radar and camera network; a system for monitoring loads generated on bollards on the berth by mooring ropes, based on digital sensors measuring the tension on bollards; a system for monitoring the use of roads, railways and waterways; and a system for monitoring water and air quality.

The Internet of Things (IOT) platform can process data collected from a large number of different sensors and models, giving them operational meaning. The IOT platform is used in most smart ports, including the Port of Rotterdam [24]. Examples of monitoring systems used in this port are networks for monitoring water depths, water levels and weather conditions; a quay wall monitoring system based on satellite survey and tilt sensors, installed on the quay walls to detect the increased risk; and a system using smart bollards for the real-time monitoring of the forces on the mooring lines.

The aim of a smart port solution, like that in the Port of Hamburg, is to achieve sustainable economic growth and maximum benefits while minimizing environmental impact [25]. IoT technology is adopted in the Port of Hamburg for monitoring infrastructure including harbor cranes and means of transport. Traffic monitoring systems cover the Port of Hamburg and Elbe approach channels.

The network of monitoring systems can be different depending on port stakeholder policies, as well as for technical and economic reasons. Its open architecture allows further development.

The example elements of an integrated monitoring system for ship operations in a seaport allowing the detection of environmental and operational limit exceedance is presented in Figure 10.

The most important and most often-used element of this system is the environmental monitoring system, which includes the current state of air, water, soil, sediments, ecosystems and habitats. The real operational data-monitoring covers the moored vessel's impact and the berthing vessel's impact on the port infrastructure, berth availability and berth accessibility.

The detection of limits includes pollution, structural component failures, damages to the berth structure, fendering system and seabed protection near the quay and the ship's hull.

The digital port solution can integrate a large amount of operational and environmental data or concentrate on optimizing logistic processes to increase the port stainability as, for instance, in the Port of Algeciras [26], with the monitoring of port operations using radars and communication system.

Lim et al. [6], who identified trends in the development of port operational sustainability by focusing on analytical sustainability indicators and increasing understanding of port

performance, concluded that the main challenges include achieving a consistent way of measuring sustainability. The authors predicted that future research should encourage the improvements in port operational practices and self-analysis to update the action priorities.

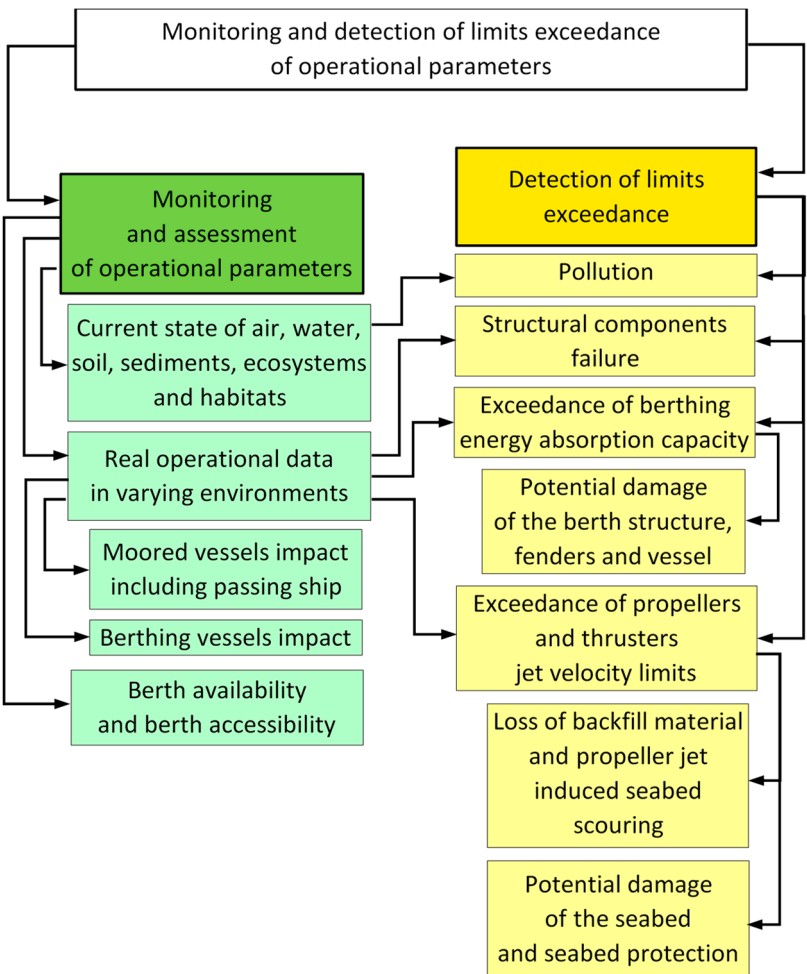

**Figure 10.** Monitoring and detection of factors related to the environment and ship maneuvering operations.

The relationships between the individual elements influencing the sustainable development of a seaport are summarized in Figure 11. The data related to the current state of air, water, soil, sediments, ecosystems and habitats, berth availability and berth accessibility in the port, along with real operational data in varying environments, can be used for establishing the analytical sustainability indicators.

The monitoring of berth availability and accessibility allows for improved planning of port operations and decreases emissions from ships. Online access to the real operational data, including the condition of berth structures and data from systems supervising large ship operation—during berthing and when the ship is moored—allows for efficient seaport management, refinement of design parameters, and validation of statistical decision-making algorithms in the raw data processing.

The proper model of ship performance is the basis for the raw data processing and analysis. Then, AI-based algorithms allow for effective tuning of the theoretical models. The virtual models of port infrastructure, port operation and online access to real environmental and operational data allow the development of digital twins of infrastructure for contemporary and future applications and, in the most advanced form, a digital twin of the entire port [13,24,27].

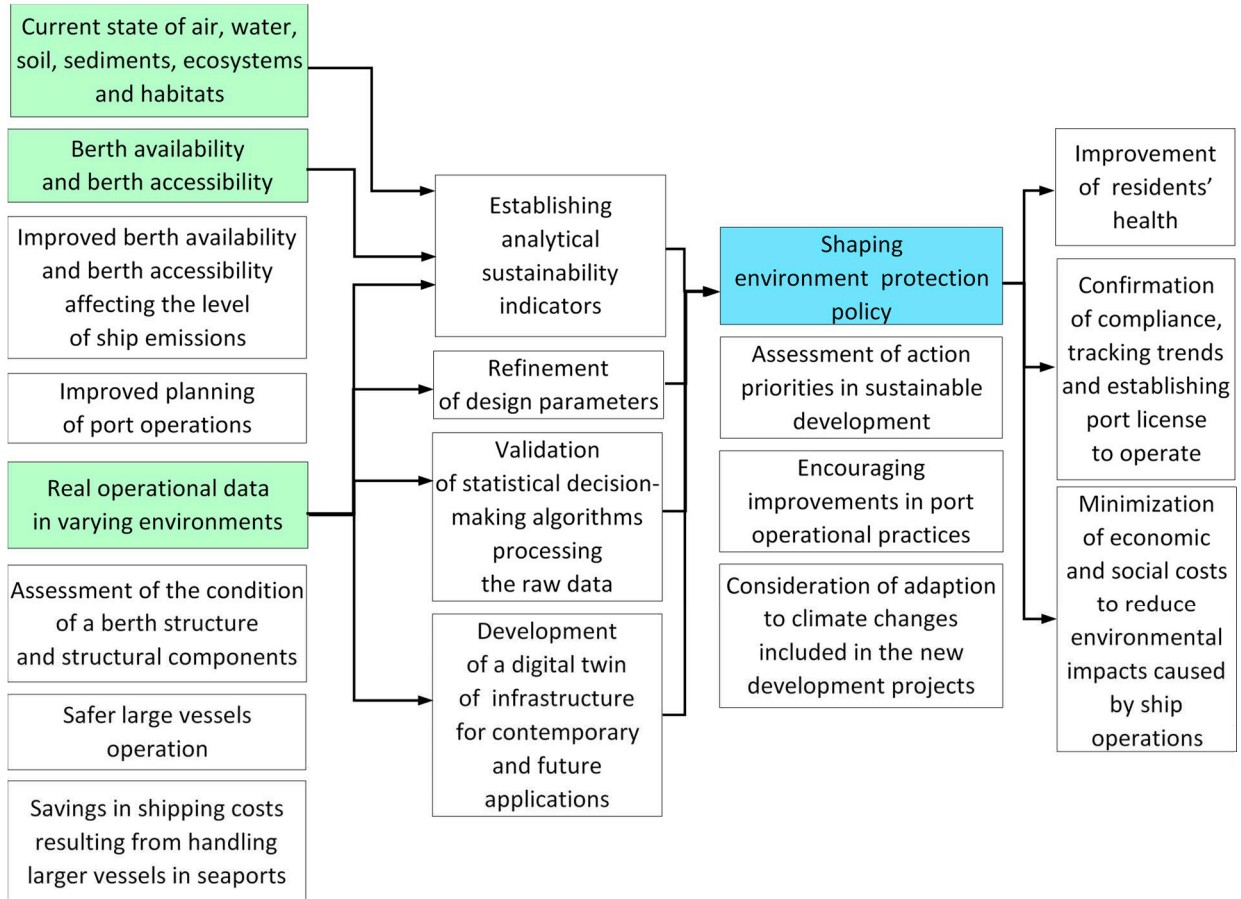

**Figure 11.** Elements influencing the sustainable development of a seaport.

The digital twin is a dynamic system that processes raw data into meaningful information encoded by the probabilistic graphical model using the statistical decision-making algorithms [28–30]. The flow from observation to decision that repeats over time can be modeled using, for example, the dynamic Bayesian network, considering important sources of uncertainty [31].

The use of the digital twin of the civil infrastructure evolves along with the acquisition and processing of the real data. The potential for its use is growing because, due to AI-based applications, the initially adopted models can be transformed into a virtual replica of the structure.

The scope of applications is expanding, and, in addition to the current assessment of the structure's condition, it allows for the following actions [32]:

1. planning emergency scenarios,
2. predicting reliability and durability of the structure,
3. planning changes in operational procedures,
4. scheduling maintenance work,
5. planning reconstruction work,
6. sequencing demolition work,
7. developing operational autonomy of infrastructures.

Shaping the environmental protection policy in the port, which functions in the intermodal logistics supply chain in the port city, is directly related to assessing priorities for sustainable development activities, encouraging the improvement of port operational practice, and taking into account adaptation to climate change in new development projects [33–37].

This means both current and lasting improvements in the health of residents, confirmation of the port's compliance with requirements, adaptation to trends in environmental

protection and the introduction of a port license for the ecological operation of ships. From an economic point of view, minimizing economic and social costs involves limiting the impact of ship operations on the environment [38–42].

In the research on seaport sustainability [3], the environmental research is focused on the pollution, the social research is mainly focused on the human resource management and the economic research is concentrated primarily on port management and borderline investment. In a wide context, it is included in the scheme of the intermodal logistics supply chain in the city.

The growing interest in smart seaport development was summarized by Pham [43], who concluded that the yearly number of scientific publications in journals, indexed on the Web of Science, has increased during the last five years by more than six times (19 scientific papers in 2022).

## 5. Conclusions

The information from the monitoring systems in seaports is important for the port services, port and ship owners and operators, as well as insurance companies, maritime safety boards and maritime courts in the case of an accident.

Real operational and environmental data can be used for validation of models of degradation processes of marine structures, better selection of design parameters and planning of maintenance work [8]. The two examples of modeling port operational procedures presented in the paper have potential when applied to port environment monitoring systems.

A smart seaport project can concentrate on the port logistics, integrate different areas of port operation and, in the final stage, become a virtual model of the entire port's digital twin. When it is decided which components should be subject to monitoring, both environmental and economic criteria are considered. The open architecture of integrated monitoring systems allows for the extension of the list of components.

**Author Contributions:** Conceptualization, T.A.-G., Z.B. and M.K.G.; methodology, T.A.-G. and Z.B.; software, T.A.-G. and J.J.; validation, T.A.-G., Z.B. and M.K.G.; formal analysis, T.A.-G.; investigation, T.A.-G., Z.B. and J.J.; resources, T.A.-G. and Z.B.; data curation, T.A.-G., Z.B. and M.K.G.; writing—original draft preparation, T.A.-G. and M.K.G.; writing—review and editing, T.A.-G., Z.B. and M.K.G.; visualization, T.A.-G. and J.J.; supervision, T.A.-G. and Z.B.; project administration, T.A.-G. and Z.B.; funding acquisition, T.A.-G. and Z.B. All authors have read and agreed to the published version of the manuscript.

**Funding:** This research was supported by the project RPPM.01.01.01-22-0068/16-00 "Development of a prototype of a system for monitoring the loads on berths and bed protection in the area of ship berthing along with the implementation of the final product on the market by Enamor Ltd. company from Gdynia" within "Smart Specializations of Pomerania Region—offshore technology, ports and logistics"—Pomeranian Voivodeship Regional Operational Program for 2014–2020 and the following academic grants from Gdynia Maritime University: WN/2023/PZ/03, WN/2023/PZ/08 and WN/PI/2023/04.

**Institutional Review Board Statement:** Not applicable.

**Informed Consent Statement:** Not applicable.

**Data Availability Statement:** The data presented in this study are available on request from the corresponding author.

**Acknowledgments:** The technical support was provided by "Prof. Lech Kobylinski Foundation for Safety of Navigation".

**Conflicts of Interest:** The authors declare no conflicts of interest. The funders had no role in the design of the study, collection, analysis and interpretation of data, writing of the manuscript or decision to publish the results.

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
