# Peer review of "Monitoring of Ship Operations in Seaport Areas in the Sustainable Development of Ocean–Land Connections"

_sustainability, doi:10.3390/su16020597_

Round 1

Reviewer 1 Report

Comments and Suggestions for Authors

The work carried out makes significant progress in a field with great potential for the future: the use of the Internet of Things (IoT) in the configuration of smart ports. I would encourage the authors in this direction.

Author Response

Dear Reviewer,

Thank you for your valuable remarks and comments that helped improve our paper. Please find the corrections.

  1. Perhaps the conclusions are a bit limited. Perhaps the highlight of the work is the field experiments and their validation

The abstract has been revised to highlight the scope of the presented research.

“Abstract: The paper is devoted towards underlining the important role of monitoring systems in the sustainable development of seaport areas—sensitive ocean-land connections exposed to the harmful effects of multimodal transport. The study concerns the existing possibilities of monitoring of the environmental factors and ship traffic near port infrastructure. The main aim of the study was presenting the example of solutions, supporting the sustainable development of port areas, related to the most dangerous ship maneuvering operations carried out near the berths. An indirect method for measuring loads on the seabed from the propeller and thruster jets during ship berthing and an experimental method for predicting the hydrodynamic forces generated on a moored Panamax size bulk carrier by a similar vessel passing along in shallow water conditions are described in the context of their implementation in monitoring systems. The cloud-based system, installed in the ferry terminal, in Port of Gdynia, developed for monitoring the flow generated by the ship propellers during maneuvers near the berth and warning about the exceedance of allowable pressure on the quay wall, allowed after two-year operation, to draw the conclusions related to maintenance planning, having impact on port sustainability. The discussion presented in the paper underlines the influence of monitoring both the environmental elements and hazardous ship operations on the sustainable development of seaport areas.”

  1. The article would benefit from other monitoring experiences in other ports, Hamburg, Algeciras, and Rotterdam are mentioned, but in a very superficial way.

The examples of monitoring systems used in the biggest European ports mentioned in the paper have been added in Discussion on pages 10, 11 and 14.

Page 10

“The monitoring systems used in Port of Antwerp are for example: traffic monitoring and management system based on digital radar and camera network; system for monitoring loads generated on bollards on the berth by mooring ropes, based on digital sensors measuring the tension on bollards; system for monitoring use of roads, railways and waterways; system for monitoring water and air quality.”

“The example monitoring systems used in this port are networks for monitoring water depths, water levels and weather conditions; quay walls monitoring system based on satellite survey and tilt sensors installed on the quay walls when the increased risk is detected; system using smart bollards for real time monitoring of the forces on the mooring lines.”

“IoT technology is adopted in Port of Hamburg for monitoring infrastructure including harbor cranes and means of transport. Traffic monitoring systems cover Port of Hamburg and Elbe approach channels.”

Page 11

“… in the Port of Algeciras [26], with the monitoring of port operations using radars and communication system.”

Page 14

“The growing interest in smart seaports development was summarized by Pham [37], who concluded that the yearly number of scientific publications in journals, indexed on Web of Science, has increased during the last five years more than six times (19 scientific papers in 2022).”

  1. Likewise, the conclusions should be more specific.

The conclusions have been extended summarizing the experimental works.

“Two examples of modeling port operational procedures presented in the paper have a potential when applied to port environment monitoring systems.”

More details summarizing the presented research are provided in  Discussion on page 10

“The fuzzy model for the determination of the maximum hydrodynamic forces generated by the passing ship on the moored ship, presented in [21] is an example of application of the presented experiments in creation of the control system for autonomous navigation. These results can be also used in traffic monitoring systems, advising the safe passing distance and speed.

The limitations of the presented study are related to the scope of the presented research including only selected cases of the wide range of port operations. The two examples of prediction and monitoring of ship operations, which have an impact on port infrastructure preventing disruptions in port operations and reducing CO2 footprint, are between different solutions used in sustainable seaports.”

  1. More less, very generic. There is a lot of experimentation and validation work here, but some contextualization is lacking.

The following text was added in Introduction on page 3 to improve the contextualization

“The structure of the presented paper is as follows: the background of the study presented in the Introduction is followed, in the second paragraph, by examples of numerical and experimental investigations carried out using large scale manned models, allowing for prediction of ship impact on port infrastructure and development of monitoring systems. In third paragraph the experimental data of passing ship effect are verified with the data available in the literature, showing the strong influence of water depth to ship draft ratio on the coefficients of hydrodynamic forces generated on the moored ship. The fourth paragraph – Discussion shows advantages of the presented investigations in case of using them in monitoring systems. Based on the authors experience, examples of monitoring systems used in the biggest European ports and literature study the influence of the monitoring of ship operations on the sustainable development of seaports is summarized. In the fifth paragraph – Conclusions the prognosis of future smart ports development is discussed.”

Reviewer 2 Report

Comments and Suggestions for Authors

This research is underling the the important role of monitoring systems in the sustainable development of seaport areas—sensitive ocean-land connections exposed to the harmful effects of multimodal transport. And the paper also proposed an indirect method to for measuring loads on the seabed at the quay from the propeller and thruster jets during ship berthing and an experimental method for predicting loads generated by a passing ship on a ship moored at the solid-type berth in shallow water conditions are presented. From the method and experimental research, this part of content is meaningful. But there is major issues as below:

The title of the paper is inconstant with the main content of the paper. You know, the title is emphasized on the monitoring of ship operations, but the main content of " Discussion" is not closed relationship to he title and the method and  experimental  research.

So I think the authors must re-organized the title and the content of the research, especially in the "Discussion" part, the author should discuss more about the method proposed is how to affect the sustainable development of ocean-land connections.

Author Response

Dear Reviewer,

Thank you for your valuable remarks and comments that helped improve our paper. Please find the corrections.

  1. The title of the paper is inconstant with the main content of the paper. You know, the title is emphasized on the monitoring of ship operations, but the main content of " Discussion" is not closed relationship to the title and the method and  experimental  research.So I think the authors must re-organized the title and the content of the research, especially in the "Discussion" part, the author should discuss more about the method proposed is how to affect the sustainable development of ocean-land connections.

The content of the research was reorganized.

Abstract has been revised according to the title.

“Abstract: The paper is devoted towards underlining the important role of monitoring systems in the sustainable development of seaport areas—sensitive ocean-land connections exposed to the harmful effects of multimodal transport. The study concerns the existing possibilities of monitoring of the environmental factors and ship traffic near port infrastructure. The main aim of the study was presenting the example of solutions, supporting the sustainable development of port areas, related to the most dangerous ship maneuvering operations carried out near the berths. An indirect method for measuring loads on the seabed from the propeller and thruster jets during ship berthing and an experimental method for predicting the hydrodynamic forces generated on a moored Panamax size bulk carrier by a similar vessel passing along in shallow water conditions are described in the context of their implementation in monitoring systems. The cloud-based system, installed in the ferry terminal, in Port of Gdynia, developed for monitoring the flow generated by the ship propellers during maneuvers near the berth and warning about the exceedance of allowable pressure on the quay wall, allowed after two-year operation, to draw the conclusions related to maintenance planning, having impact on port sustainability. The discussion presented in the paper underlines the influence of monitoring both the environmental elements and hazardous ship operations on the sustainable development of seaport areas.”

The structure of the paper has been described in Introduction on page 3

“The structure of the presented paper is as follows: the background of the study presented in the Introduction is followed, in the second paragraph, by examples of numerical and experimental investigations carried out using large scale manned models, allowing for prediction of ship impact on port infrastructure and development of monitoring systems. In third paragraph the experimental data of passing ship effect are verified with the data available in the literature, showing the strong influence of water depth to ship draft ratio on the coefficients of hydrodynamic forces generated on the moored ship. The fourth paragraph – Discussion shows advantages of the presented investigations in case of using them in monitoring systems. Based on the authors experience, examples of monitoring systems used in the biggest European ports and literature study the influence of the monitoring of ship operations on the sustainable development of seaports is summarized. In the fifth paragraph – Conclusions the prognosis of future smart ports development is discussed.”

The details summarizing the presented research are provided in  Discussion on page 10

“The fuzzy model for the determination of the maximum hydrodynamic forces generated by the passing ship on the moored ship, presented in [21] is an example of application of the presented experiments in creation of the control system for autonomous navigation. These results can be also used in traffic monitoring systems, advising the safe passing distance and speed. The limitations of the presented study are related to the scope of the presented research including only selected cases of the wide range of port operations. The two examples of prediction and monitoring of ship operations, which have an impact on port infrastructure preventing disruptions in port operations and reducing CO2 footprint, are between different solutions used in sustainable seaports.”

The conclusions have been extended summarizing the experimental works.

“Two examples of modeling port operational procedures presented in the paper have a potential when applied to port environment monitoring systems.”

Reviewer 3 Report

Comments and Suggestions for Authors

The manuscript deals titled ’Monitoring of ship operations in seaport areas in the sustainable development of ocean-land connections’ with a scientific and current issue. The manuscript has been structured adequately, it thus is understandable and clear. The final version of the paper is not necessary to modify and can be published in the journal Sustainability. 

Nice work! 

Author Response

Dear Reviewer,

Thank you for your kind remarks and comments.

Reviewer 4 Report

Comments and Suggestions for Authors

The manuscript considers the monitoring system environmental factors and ship traffic near port infrastructure. The results of experimental studies and model tests are compared to the existing empirical methods. The paper discussed the cloud-based monitoring allowing introduction of digital twins of marine constructions in the port. The topic is relevant, it address a specific gap in the field. Nonetheless, there are aspects that require improvements:

1) In the first section, the authors should more clearly define the motivation, and contribution of the research, as well as the organization of this paper.

2) The presented architecture of the cloud monitoring system (Fig. 4) requires additional explanations. Purpose of functional modules, types of interaction, hardware and software characteristics, etc.

3) The authors refer to the fuzzy approach [21], but do not draw conclusions about the possibility of its application in this work.

4) The discussion of the results should clearly show the reader the main limitations of the proposed approach.

Author Response

Dear Reviewer,

Thank you for your valuable remarks and comments that helped improve the paper. Please find our corrections.

1. In the first section, the authors should more clearly define the motivation, and contribution of the research, as well as the organization of this paper.

The abstract has been revised to show the motivation and contribution of the research

“Abstract: The paper is devoted towards underlining the important role of monitoring systems in the sustainable development of seaport areas—sensitive ocean-land connections exposed to the harmful effects of multimodal transport. The study concerns the existing possibilities of monitoring of the environmental factors and ship traffic near port infrastructure. The main aim of the study was presenting the example of solutions, supporting the sustainable development of port areas, related to the most dangerous ship maneuvering operations carried out near the berths. An indirect method for measuring loads on the seabed from the propeller and thruster jets during ship berthing and an experimental method for predicting the hydrodynamic forces generated on a moored Panamax size bulk carrier by a similar vessel passing along in shallow water conditions are described in the context of their implementation in monitoring systems. The cloud-based system, installed in the ferry terminal, in Port of Gdynia, developed for monitoring the flow generated by the ship propellers during maneuvers near the berth and warning about the exceedance of allowable pressure on the quay wall, allowed after two-year operation, to draw the conclusions related to maintenance planning, having impact on port sustainability. The discussion presented in the paper underlines the influence of monitoring both the environmental elements and hazardous ship operations on the sustainable development of seaport areas.”

The following text was added in Introduction on page 3 to improve the contextualization:

 “The structure of the presented paper is as follows: the background of the study presented in the Introduction is followed, in the second paragraph, by examples of numerical and experimental investigations carried out using large scale manned models, allowing for prediction of ship impact on port infrastructure and development of monitoring systems. In third paragraph the experimental data of passing ship effect are verified with the data available in the literature, showing the strong influence of water depth to ship draft ratio on the coefficients of hydrodynamic forces generated on the moored ship. The fourth paragraph – Discussion shows advantages of the presented investigations in case of using them in monitoring systems. Based on the authors experience, examples of monitoring systems used in the biggest European ports and literature study the influence of the monitoring of ship operations on the sustainable development of seaports is summarized. In the fifth paragraph – Conclusions the prognosis of future smart ports development is discussed.”

2. The presented architecture of the cloud monitoring system (Fig. 4) requires additional explanations. Purpose of functional modules, types of interaction, hardware and software characteristics, etc.

The system is described in reference [8]. The following text is added on page 4

“The functional modules, hardware and software characteristics of the system are described in [8].”

3. The authors refer to the fuzzy approach [21], but do not draw conclusions about the possibility of its application in this work.

The part of text related to the fuzzy approach was rearranged and moved to Discussion on page 10.

 “The fuzzy model for the determination of the maximum hydrodynamic forces generated by the passing ship on the moored ship, presented in [21] is an example of an application of the presented experiments in creation of the control system for autonomous navigation. These results can be also used in traffic monitoring systems, advising the safe passing distance and speed.”

4. The discussion of the results should clearly show the reader the main limitations of the proposed approach.

The following text related to thelimitations is added in Discussion on page 10

“The limitations of the presented study are related to the scope of the presented research including only the selected cases of the wide range of port operations. Two examples of prediction and monitoring of the ship operations, which have an impact on the port infrastructure preventing disruptions in port operation and reducing CO2 footprint, are between different solutions used in sustainable seaports.”

Reviewer 5 Report

Comments and Suggestions for Authors

Manuscript: Monitoring of ship operations in seaport areas in the sustainable development of ocean-land connections.

Please consider the following:

Abstract (even if some aspects are included, more specific elaboration is required), please revise the abstract, starting from clear purpose (aims) of the study, then addressing research scope and method and completing with brief results.

Discussion: it is not clear if Figure 10. Monitoring and detection of factors related to the environment and ship manoeuvring and Figure 11. Monitoring and detection of factors related to the environment and ship manoeuvring are elaborated/modified by Authors or copied from other sources. Please state clearly what is your input into the schemes mentioned above.

Conclusions; must be strengthen and  referenced in greater extend  to the study results.

 Summing-up, the manuscript content requires some adjustments in Abstract, Discussion and Conclusions and if the concerns will be resolved, the manuscript can be considered for publication

Comments on the Quality of English Language

Well written, no remarks

Author Response

Dear Reviewer,

Thank you for your valuable remarks and comments that helped improve our paper. Please find the corrections of Abstract, Discussion and Conclusions.

  1. Abstract (even if some aspects are included, more specific elaboration is required), please revise the abstract, starting from clear purpose (aims) of the study, then addressing research scope and method and completing with brief results.

The abstract has been revised

“Abstract: The paper is devoted towards underlining the important role of monitoring systems in the sustainable development of seaport areas—sensitive ocean-land connections exposed to the harmful effects of multimodal transport. The study concerns the existing possibilities of monitoring of the environmental factors and ship traffic near port infrastructure. The main aim of the study was presenting the example of solutions, supporting the sustainable development of port areas, related to the most dangerous ship maneuvering operations carried out near the berths. An indirect method for measuring loads on the seabed from the propeller and thruster jets during ship berthing and an experimental method for predicting the hydrodynamic forces generated on a moored Panamax size bulk carrier by a similar vessel passing along in shallow water conditions are described in the context of their implementation in monitoring systems. The cloud-based system, installed in the ferry terminal, in Port of Gdynia, developed for monitoring the flow generated by the ship propellers during maneuvers near the berth and warning about the exceedance of allowable pressure on the quay wall, allowed after two-year operation, to draw the conclusions related to maintenance planning, having impact on port sustainability. The discussion presented in the paper underlines the influence of monitoring both the environmental elements and hazardous ship operations on the sustainable development of seaport areas.”

  1. Discussion: it is not clear if Figure 10. Monitoring and detection of factors related to the environment and ship manoeuvring and Figure 11. Monitoring and detection of factors related to the environment and ship manoeuvring are elaborated/modified by Authors or copied from other sources. Please state clearly what is your input into the schemes mentioned above.

The figures 10 and 11 were elaborated by the authors, based on their own experience and literature.

  1. Conclusions; must be strengthen and referenced in greater extend to the study results.

More details summarizing the presented research are provided in the Discussion on page 10

“The fuzzy model for the determination of the maximum hydrodynamic forces generated by the passing ship on the moored ship, presented in [21] is an example of an application of the presented experiments in creation of the control system for autonomous navigation. These results can be also used in traffic monitoring systems, advising the safe passing distance and speed.

“The limitations of the presented study are related to the scope of the presented research including only the selected cases of the wide range of port operations. Two examples of prediction and monitoring of the ship operations, which have an impact on the port infrastructure preventing disruptions in port operation and reducing CO2 footprint, are between different solutions used in sustainable seaports.”

Conclusions have been extended summarizing the presented study.

“Two examples of modeling port operational procedures presented in the paper have a potential when applied to port environment monitoring systems.”

Round 2

Reviewer 2 Report

Comments and Suggestions for Authors

Thanks for authors revision. I think the paper now can be accepted in present form!